# A rule-based multiscale model of hepatic stellate cell plasticity: Critical role of the inactivation loop in fibrosis progression

**Matthieu Bouguéon**[1,2], **Vincent Legagneux**[2], **Octave Hazard**[3,4], **Jérémy Bomo**[1,2], **Anne Siegel**[1], **Jérôme Feret**[4,5] *, **Nathalie Théret**[1,2] *

**1** Univ Rennes, Inria, CNRS, IRISA, UMR 6074, Rennes, France, **2** Univ Rennes, Inserm, EHESP, Irset, UMR S1085, Rennes, France, **3** École Polytechnique, Palaiseau, France, **4** DI-ENS (Inria, ÉNS, CNRS, PSL University), École normale supérieure, Paris, France, **5** Team Antique, Inria, Paris, France

* jerome.feret@ens.fr (JF); nathalie.theret@inserm.fr (NT)

**Data Availability Statement:** The 3 model family files are available in the GitHub repository: https://github.com/MBougueon/HSC_model_Kappa.

## Abstract

Hepatic stellate cells (HSC) are the source of extracellular matrix (ECM) whose overproduction leads to fibrosis, a condition that impairs liver functions in chronic liver diseases. Understanding the dynamics of HSCs will provide insights needed to develop new therapeutic approaches. Few models of hepatic fibrosis have been proposed, and none of them include the heterogeneity of HSC phenotypes recently highlighted by single-cell RNA sequencing analyses. Here, we developed rule-based models to study HSC dynamics during fibrosis progression and reversion. We used the Kappa graph rewriting language, for which we used tokens and counters to overcome temporal explosion. HSCs are modeled as agents that present seven physiological cellular states and that interact with (TGF$\beta$1) molecules which regulate HSC activation and the secretion of type I collagen, the main component of the ECM. Simulation studies revealed the critical role of the HSC inactivation process during fibrosis progression and reversion. While inactivation allows elimination of activated HSCs during reversion steps, reactivation loops of inactivated HSCs (iHSCs) are required to sustain fibrosis. Furthermore, we demonstrated the model's sensitivity to (TGF$\beta$1) parameters, suggesting its adaptability to a variety of pathophysiological conditions for which levels of (TGF$\beta$1) production associated with the inflammatory response differ. Using new experimental data from a mouse model of CCl4-induced liver fibrosis, we validated the predicted ECM dynamics. Our model also predicts the accumulation of iHSCs during chronic liver disease. By analyzing RNA sequencing data from patients with non-alcoholic steatohepatitis (NASH) associated with liver fibrosis, we confirmed this accumulation, identifying iHSCs as novel markers of fibrosis progression. Overall, our study provides the first model of HSC dynamics in chronic liver disease that can be used to explore the regulatory role of iHSCs in liver homeostasis. Moreover, our model can also be generalized to fibroblasts during repair and fibrosis in other tissues.

Using the KAPPA user interface downloadable from https://kappalanguage.org/download, the simulations can be performed by uploading the file of the model family reactMFB-with-inactivation.ka. We used the version 4.2.12 of Kappa and 4.13.1 of OCaml (required to run Kappa). The python scripts use to launch parallelized simulation are available in the GitHub repository: https://github.com/MBougueon/simulation_pipeline.

**Funding:** This work was supported by the Institut National de la Sante et de la Recherche Medicale (Inserm)(NT, VL,MB), the Institut national de recherche en informatique et en automatique (INRIA)(JF), the Centre National de la Recherche Scientifique (AS) and the University of Rennes 1 (JB). The funders had no role in study design, data collection and analysis, decision to publish, or preparation of the manuscript.

**Competing interests:** The authors have declared that no competing interests exist.

## Author summary

Chronic liver diseases (CLDs) are associated with the development of fibrosis which is characterized by an abnormal deposition of extracellular matrix (ECM) leading to severe liver dysfunction. Hepatic stellate cells (HSCs) are key players in liver fibrosis driving ECM remodeling. However numerous biological processes are involved including HSC activation, proliferation, differentiation and inactivation and novel computational modeling is necessary to integrate such complex dynamics. Here, we used the Kappa graph rewriting language to develop the first rule-based model describing the HSCs dynamics during liver fibrosis and its reversion. Simulation analyses enabled us to demonstrate the critical role of the HSC inactivation loop in the development of liver fibrosis, and to identify inactivated HSCs as potential new markers of fibrosis progression.

## Introduction

Liver fibrosis is an excessive wound healing response induced by chronic injuries, mainly caused by viral hepatitis (HCV, HBV), alcohol abuse and non-alcoholic steatohepatitis (NASH). Fibrosis is characterized by an accumulation of extracellular matrix (ECM) which increases the stiffness of tissues, leading to severe liver dysfunction. The final stage of fibrosis, cirrhosis, leads to complications such as ascites, variceal hemorrhage, encephalopathy and hepatocellular carcinoma, and is associated with a high mortality rate worldwide [1]. Activation of hepatic stellate cells (HSCs) is the main process underlying hepatic fibrosis [2]. In a normal liver, HSCs are quiescent, store vitamin A and are located in the Disse space between hepatocytes and endothelial cells that delineate sinusoids. Upon liver injury, HSCs are activated and transdifferentiate into ECM-secreting myofibroblasts (MFB) that contribute to tissue repair. In chronic liver disease, the inflammatory signal persists, leading to sustained HSC activation, ECM accumulation and fibrosis [2–4]. Importantly, reversibility or regression of liver fibrosis after elimination of the inflammatory agent has been observed in experimental models and clinical studies [5, 6]. This phenomenon is associated with the elimination of myofibroblasts, involving processes of apoptosis, senescence and inactivation, but the contribution of each of these mechanisms to the repair/fibrosis balance remains unclear.

Understanding the behavior of HSCs during liver injuries requires multiscale modeling approaches. Such approaches have been widely developed for modeling biological processes [7] and diseases [8, 9]. Because fibrosis is a pathological tissue repair activity that occurs in different organs, unified approaches have been proposed by including common components such as the inflammatory response and extracellular matrix remodeling [10]. However, each tissue is characterized by specific cell microenvironments. Different multiscale models have therefore been developed for cardiac [11] and pulmonary [12] fibrosis, and various approaches have been used to model the progression of liver fibrosis. Among them, the agent-based (ABM) multiscale model developed by Dutta-Moscato *et al.* [13] successfully reproduced the experimental collagen deposition observed in carbon tetrachloride (CCl4)-induced fibrosis in rats. The model integrated parenchymal cells, inflammatory cells, collagen-producing cells and molecular regulatory agents, with rules defining the properties and interactions between all agents. This model was then modified by preventing the migration of collagen-producing cells, thus providing a more accurate dynamic of collagen deposition [14]. The Dutta-Moscato model has also been extended and modified by Wand and Jiang [15] to include information about lipid accumulation induced by CCl4 treatment, making it possible to study the progression of liver fibrosis in presence or absence of steatosis. In addition to these agent-based

models, Friedman and Hao recently published a partial differential equation (PDE) model for liver fibrosis that includes information on inflammation regulation and ECM remodeling, enabling exploration of anti-fibrotic drugs [16]. Although instructive, all these models reduce the dynamics of HSCs to their activation to become ECM-producing myofibroblasts, without taking into account the plasticity of HSCs recently illustrated by large-scale single-cell analyses. [17–19]. Among the complex behaviors of HSCs, the authors identified heterogenous cell phenotypes such as proliferating cells and ECM-producing cells. In addition the inactivation processes remain poorly understood and information is lacking about the fate of inactivated HSCs (iHSCs) during liver fibrosis and reversion. The present study aims to decipher HSC dynamics by developing multiscale models which integrate for the first time all the HSC states during the development of fibrosis and its reversion. The multi-state combinatorial nature of the cells studied led us to choose a rule-based model (RBM) approach, much more appropriate for dealing with this type of network. Among these formalisms, we selected the Kappa language [20–22]. Using the Kappa language, our model describes the interactions between all cell populations and molecules such as transforming growth factor $\beta$ 1 (TGF$\beta$1) which induces HSC activation and promotes secretion of type I collagen, the major component of ECM [23]. We validated the predicted ECM dynamics using new experimental data from a mouse model of CCl4 induced liver fibrosis. Importantly, simulation studies demonstrated the regulatory role of HSC inactivation loops during liver fibrosis and predicted the accumulation of inactivated HSCs during chronic liver disease. By analyzing RNA sequencing data from 102 patients with non-alcoholic steatohepatitis (NASH) associated with liver fibrosis, we confirmed this accumulation, identifying iHSCs as novel markers of fibrosis progression.

## Results

### Three rule-based model families for HSCs dynamics

We integrated information based on literature, to describe HSC dynamics during liver fibrosis and its reversion however the lack of some information led us to propose hypotheses and finally to develop three different families of models. As shown in Fig 1A, the three families share common processes involving the different states of HSCs regulated by TGF$\beta$1. In normal liver, HSCs are maintained in a non-proliferative quiescent state (qHSC) and store vitamin A. Upon liver injury (virus, toxic, etc.), the inflammatory response leads to the production of TGF$\beta$1 which activates qHSCs into an activated state (aHSC). The fully activated HSCs are characterized by a myofibroblast (MFB) phenotype. Upon removal of liver injury, i.e in the absence of TGF$\beta$1 in the models, MFBs are either eliminated through apoptosis and senescence pathways (apop_sene_MFB) or reverted to an inactivated HSC state (iHSC) close to but different from the quiescent state. Upon a new liver injury, i.e in the presence of TGF$\beta$1 in the models, the iHSCs are reactivated (react_HSCs). In the same way that HSCs transform into MFBs, react_HSCs transform into reactivated MFBs (react_MFBs) however information is lacking about the fate of these cells. To overcome this issue, we developed two families of models in which the react_MFBs are either i) completely eliminated by the apoptosis/senescence pathways but not by inactivation, this family of models called *reactMFB-wo-inactivation* (wo for without) is described in Fig 1A$_1$), or ii) eliminated in the same way as MFBs, i.e. by the apoptosis/senescence pathways and/or the inactivation pathway, this family of models called *reactMFB-with-inactivation* is described in Fig 1A$_2$).

Similarly to react_MFB, information is lacking about the fate of iHSCs. While the activation of iHSCs by TGF$\beta$1 has been described using *in vivo* and *in vitro* experiments [24, 25], no data have been reported about the fate of iHSCs in the absence of TGF$\beta$1. However, Kisseleva *et al.* [24] showed a decrease in the iHSC population during the reversion process. We therefore

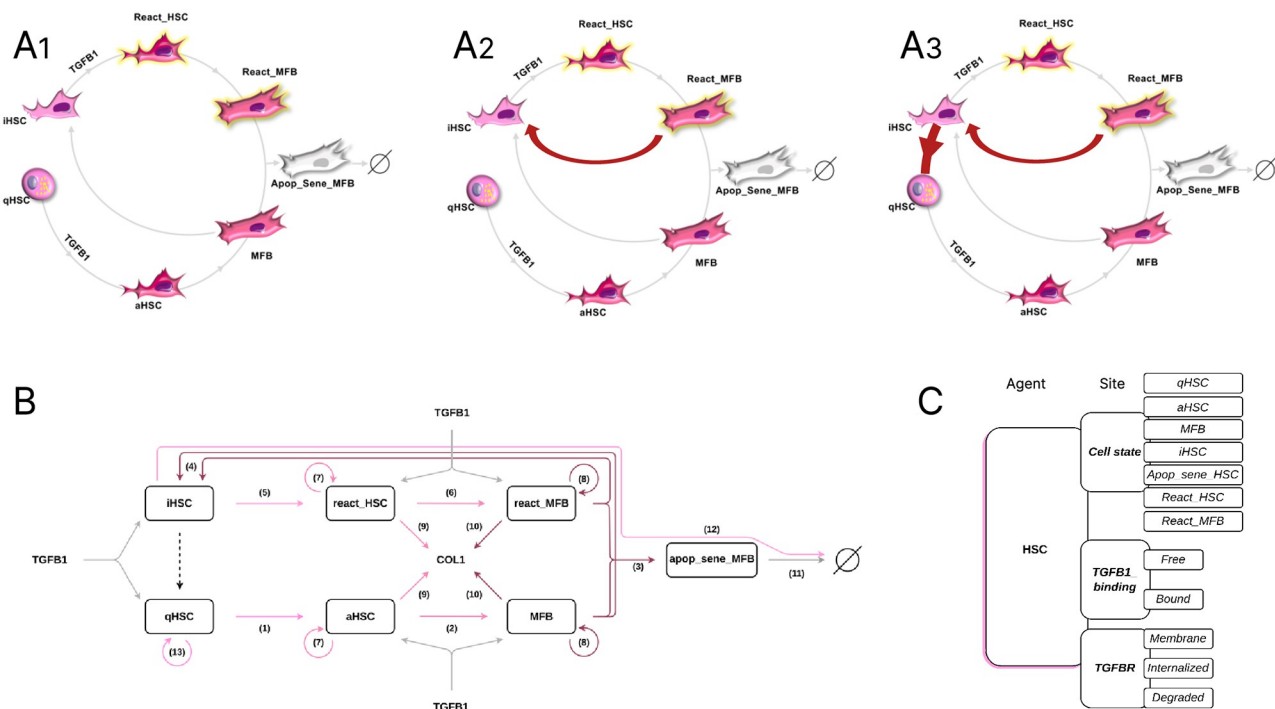

**Fig 1. Three models to describe HSC dynamics. A1** Schematic representation of models *reactMFB-wo-inactivation*; **A2** Models *reactMFB-with-inactivation* and **A3** Models *iHSC-reversion-to-qHSC*. (**B**) Schematic representation of Kappa-modeled processes in all models. HSCs are agents characterized by seven cell physiological states (qHSC, aHSC, MFB, iHSC, apop_sene_MFB, react-HSC, and react_MFB). Rules describe the processes: (1) qHSCs are activated into aHSCs, (2) aHSCs are transformed into MFBs, (3) MFBs are transformed into apop_sene_MFBs to be eliminated, (4) MFBs are inactivated into iHSCs, (5) iHSCs are reactivated into react_HSCs upon TGFβ1 stimulation, (6) react_HSCs are transformed into react_MFBs. Similarly to MFBs, react_MFBs are eliminated by apoptosis and senescence (3) or by inactivation (4). (7) aHSCs and react_HSCs proliferate, (8) MFBs and react_MFBs proliferate, (9) aHSCs and react_HSCs produce COL1, (10) MFBs and react_MFBs produce COL1, (11) apop_sene_MFBs are degraded, (12) iHSCs are degraded. (13) qHSCs self-renewal. The effect of TGFβ1 is represented by gray arrows, it induces the activation of qHSCs, the reactivation of iHSCs, and the production of COL1 in aHSCs, react_HSCs, MFBs and react_MFBs. The reversion of iHSCs into qHSCs are represented by a dotted arrow. (**C**) Kappa contact map. HSCs are agents with three sites: *cell_state* (qHSC, aHSC, MFB, iHSC, react_HSC, react_MFB, apop_sene_MFB), *TGFB1_binding* (free or bound) and *TGFBR* (membrane, internalized or degraded).

developed a third family of models in which, in the absence of TGFβ1, iHSCs are either eliminated or return to a quiescent state, thereby participating in the repopulation of the injured liver with new qHSCs. This potential new source of qHSCs needs to be balanced with the self-renewal of native qHSCs in order to restore HSC homeostasis in the repaired liver. This models in which iHSCs revert to a quiescent state constitutes the third family called *iHSC-reversion-to-qHSC* which is illustrated in Fig 1A3.

Finally, we developed three families of models: *reactMFB-wo-inactivation*, *reactMFB-with-inactivation*, *iHSC-reversion-to-qHSC*. The biological processes of all the models have been integrated in Fig 1B, where the interactions between TGFβ1 and aHSC, react_HSC, MFB and react_MFB cells leading to the induction of their proliferation and the production of COL1 are indicated.

The three families of models were built using the Kappa rule-based language (see Methods section) and simulations were performed using the KaSim tool [26] based on the Gillespie stochastic simulation algorithm (SSA) [27]. In these Kappa models, the hepatic stellate cells are the agents which are characterized by three sites (Fig 1C) which are: i) the *cell_state* of HSCs (qHSC, aHSC, MFB, iHSC, react_HSC, react_MFB, apop_sene_MFB), ii) the *TGFB1_binding* that can be free or bound and iii) the state of the TGFβ1 receptor (*TGFBR*) that can be

localized at the membrane, internalized or degraded. In addition to these agents, we introduced *counters* to scale the intermediate steps between cell states, and we used *tokens* to describe TGF$\beta$1 and COL1 quantities instead of detailing the behavior of each molecule, thereby highly reducing the computational cost (see Methods section).

The *reactMFB-wo-inactivation* models contain 75 rules and 41 parameters. A rule for react_MFB inactivation and two parameters controlling the proportion between inactivated and eliminated react_MFB have been added to obtain the *reactMFB-with-inactivation* models, containing 76 rules and 43 parameters. An additional rule for iHSC reversion and two other parameters controlling the proportion between reverted and eliminated iHSC have been added to obtain the *iHSC-reversion-to-qHSC* models. These rules are grouped into 7 biological processes: 1) qHSC renewal, 2) binding of TGF$\beta$1 to HSCs, 3) TGFBR dynamics, 4) HSC activation and differentiation, 5) HSC proliferation, 6) COL1 production and elimination and 7) MFB inactivation (detailed in Methods section).

## Inactivation loops of react_MFBs are essential to maintain COL1 accumulation during chronic liver injury

To identify the families of models that predict HSC behaviors and collagen accumulation that fits with the experimental data from Kiesseleva *et al.* [24], we conducted simulation studies using the protocol described by the authors for a mouse model of liver fibrosis and reversion, i.e. twice-weekly CCl4 (TGF$\beta$1 in the model) injections for 8 weeks and an overall reversion period of six months. Observations reported by Kisseleva *et al.* [24] included a 1.43-fold increase in cells expressing $\alpha$-SMA and a $\sim$ 14-fold increase in the percentage of COL1 deposits at 8 weeks and a return to initial values at 6 months after the last injection. In addition, the authors assessed iHSCs/qHSCs ratios and identified a 50/50 distribution at one month and a total number of HSCs similar to initial conditions at six months of reversion. As shown in Fig 2, most qHSCs (98%) were immediately activated upon the first stimulation of TGF$\beta$1 across all models, a small number (2±1%) of qHSCs persisting during the stimulation period. Activation of qHSCs led to the production of aHSCs that differentiated into MFBs. These cells express $\alpha$-SMA protein, and we tracked $\alpha$-SMA positive cells by summing the occurrences of aHSCs, MFBs, react_HSCs and react_MFBs. Note that even if the number of qHSCs and iHSCs is low, they can be activated or reactivated by TGF$\beta$1 during repeated stimulation for two months. In line with this observation, Rosenthal *et al.* [28] recently suggested that iHSCs remain at a low level even during the activation phase. Comparison of the models showed that the react_MFB inactivation loop greatly affects model behavior. When inactivation of react_MFBs was not allowed (models *reactMFB-wo-inactivation*)(Fig 2A), we observed a lower amount of $\alpha$-SMA expressing cells (0.8-fold±0.02) and of COL1 deposits (9.5-fold±0.16) than expected at 8 weeks, with amounts even decreasing before stimulation was stopped. Conversely, when inactivation of react_MFBs was allowed (models *reactMFB-with-inactivation*) (Fig 2B, 2C and 2D) we observed an accumulation of $\alpha$-SMA expressing cells and COL1 deposits that varied according to the percentage of react_MFB inactivation. While 5% allows the model to fit with experimental observations (red circle in Fig 2), increasing the percentage to 10 and 50% led to higher amounts of $\alpha$-SMA expressing cells (1.6-±0.05 and 5-fold±0.03, respectively) and COL1 deposits (16-±0.36 and 40-fold±0.16, respectively) at 8 weeks. Only the model with 5% of react_MFB inactivation led to a 50/50 distribution of the iHSCs/qHSCs ratio at one month and a return to the initial total number of cells at six months of reversion, as described in the experimental data [24].

Based on the models *reactMFB-with-inactivation* parameterized with 5% of react_MFB inactivation, we next evaluated the impact of a potential reversion of iHSCs to a quiescence

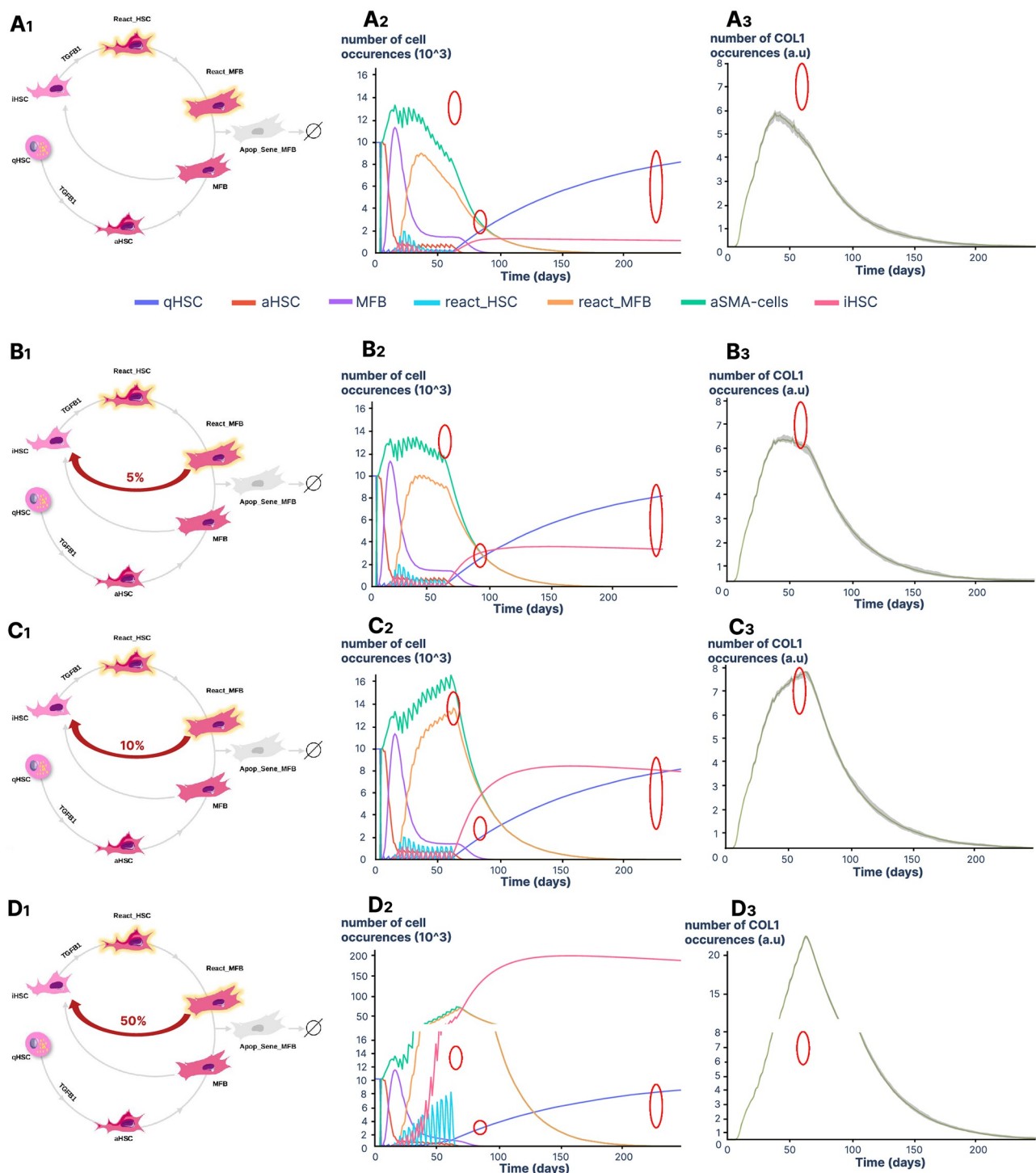

**Fig 2. Time-course analyses of cells and COL1 in models without or with react_MFB inactivation.** A series of simulations was performed using conditions of stimulation reported in Kisseleva *et al.* [24]. We used 10,000 TGFβ1 molecules per cell and 16 stimulations (twice a week) during 2 months. **A1** Models *reactMFB-wo-inactivation.* **B1** Models *reactMFB-with-inactivation* in which 5% of react_MFBs are inactivated. **C1** Models *reactMFB-with-inactivation* in which 10% of react_MFBs are inactivated. **D1** Models *reactMFB-with-inactivation* in which 50% of react_MFBs are inactivated. Stimulation of cells with TGFβ1 started on day 4 to allow the model to equilibrate. **A2,B2,C2** and **D2** display the variation in the number of cell occurrences for qHSCs, aHSCs, MFBs, iHSCs, react_HSCs, react_MFBs and α-SMA positive cells. The number of α-SMA cells is the sum of the number of aHSCs, MFBs, react_HSCs and react_MFBs. Simulations are expressed as the mean of 10 replicates. **A3**, **B3**, **C3** and **D3** display the variation of COL1% area. Simulations are expressed as arbitrary unit (a.u) and all 10 replicates are represented. The red circles indicate the biological observations reported in [24]. From left to right (**A2,B2,C2** and **D2**), the first circle is for an 1.43-fold increase in cells expressing α-SMA at 8 weeks, the

second circle is for an equal proportion of iHSCs and qHSCs at one month of recovery and the third circle is for an equal number of cells at 6 months and of cells at T0. The circle for COL1 simulations (**A3**, **B3**, **C3** and **D3**) is for a $\sim$ 14-fold increase in the percentage of COL1 deposits at 8 weeks.

state. As described in these new models *iHSC-reversion-to-qHSC*, we observed that regardless of the percentage of reversible cells, the overall behavior of the other model components was not affected (see S1 Fig).This may be due mainly to the slow dynamics of iHSCs elimination and quiescent cell renewal, which limit the impact of reversion. We have discarded this model, which was developed on the hypothesis of iHSC reversion but without any supporting experimental observations.

Taken together, our data demonstrate that inactivation of react_MFBs is required to sustain COL1 accumulation by maintaining the level of $\alpha$-SMA expressing cells, thus validating the models *reactMFB-with-inactivation*. Simulation analyses allowed us to determine that a 5% ratio between reac_MFB inactivation and apoptosis_senescence pathways reflected the cells and COL1 behaviors seen experimentally. Regarding iHSCs, our observations suggest that complete reversion of iHSCs to a quiescent state is a potential pathway to eliminate iHSC in the absence of TGF$\beta$1. Since the outcome generated by the models *iHSC-reversion-to-qHSC*, in which iHSCs can revert to qHSCs was similar to the one obtained in the models *reactMFB-with-inactivation* (S1 Fig) we chose to retain the latter for the rest of the analyses.

## Collagen I accumulation is sensitive to the dynamics of TGF$\beta$1 stimulation

Our models are calibrated using experimental data from a mouse model of CCl4-induced fibrosis [24], a reproducible experimental model widely used to study fibrosis and repair [29]. Although the protocols vary somewhat in terms of the number, frequency and modalities of CCl4 injections, these experimental models induce similar liver fibrosis. Other experimental models of induction of liver fibrosis in rodents have been developed, including injection of other chemical agents (such as thioacetamide (TAA) and dimethylnitrosamine (DMN)), high-fat diets and surgical interventions (such as bile duct ligation) [30, 31]. Whatever the etiology, induced damages lead to TGF$\beta$1-dependent activation of HSCs, but with variable dynamics. It can happen very quickly in a model induced by CCl4 or slower in a model induced by a fatty diet. To explore the sensitivity of our validated model *reactMFB-with-inactivation*, we varied the concentration of TGF$\beta$1 and the number and periodicity of stimulations. As shown in Fig 3, the model was highly sensitive to the amount of TGF$\beta$1, but this sensitivity varied with the number of stimuli and periodicity, particularly for TGF$\beta$1 values lower than 10,000 molecules per cell. Beyond that, the number of molecules exceeded the number of receptors per cell, and the model became saturated, as shown by the superposition of kinetic curves at 10,000 (green) and 100,000 (yellow) molecules per cell. At lower TGF$\beta$1 levels, collagen accumulation was much more subtle. When stimulations were spaced sufficiently far apart, collagen levels returned to baseline and, remarkably, we observed a much stronger induction upon second stimulation, particularly at doses of 10,000 and 100,000 molecules per cell (Fig 3A, periodicity 30 and 60 days). This induction of collagen 1 was less pronounced at low doses, but increasing the number of stimuli (Fig 3B, periodicity 180 hours and 15 days) showed that collagen continues to accumulate with each new addition of TGF$\beta$1 at the dose of 1000 molecules per cell, whereas it decreased for doses 10,000 and 100,000 molecules per cell. This effect remained visible for a larger number of stimulations (Fig 3C, periodicity 180h and 15 days and Fig 3D, periodicity 90h and 180h).

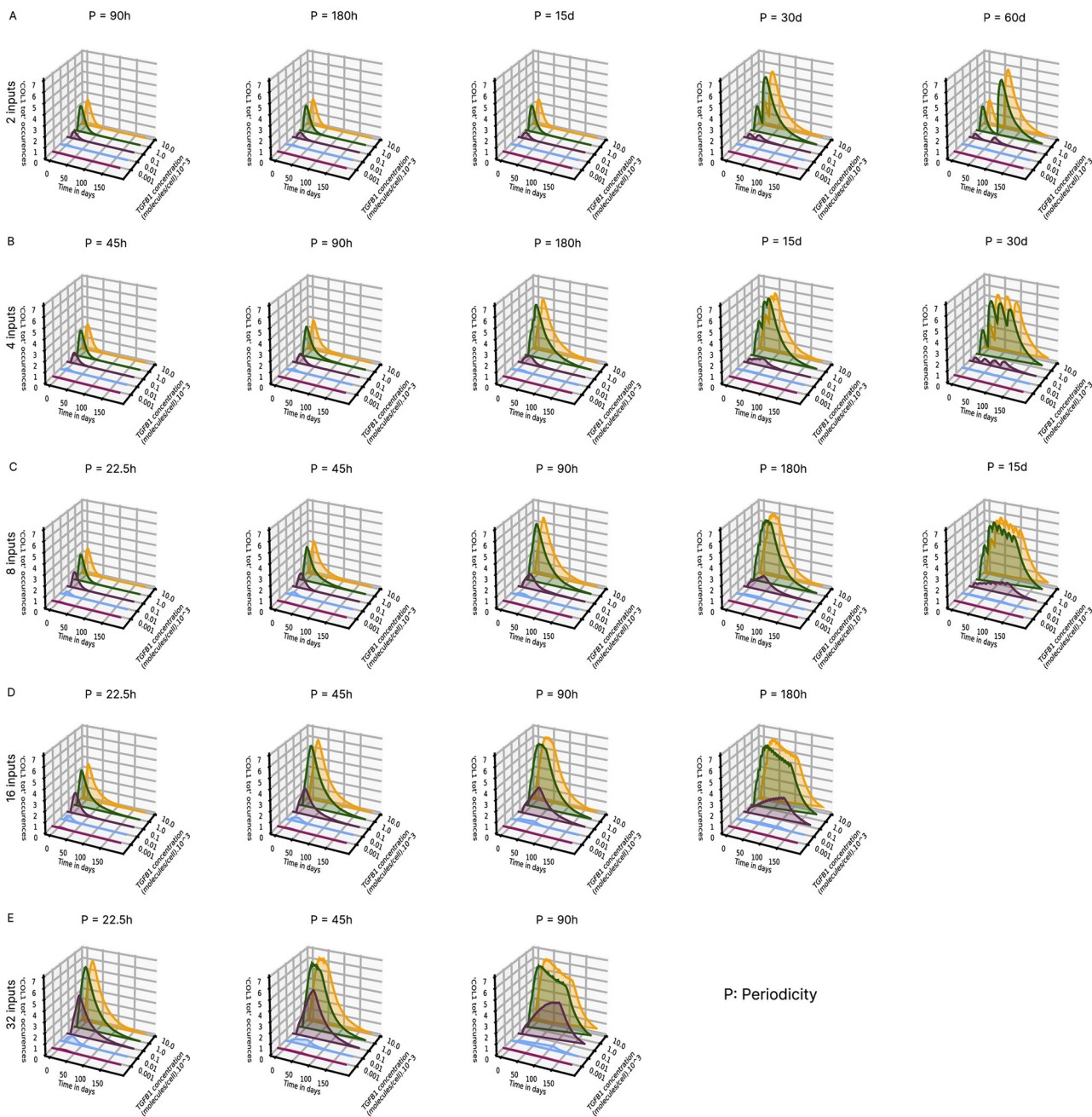

**Fig 3. Time course analysis of COL1 as a function of the variation in TGFβ1 concentration, number and periodicity of stimuli.** Results are expressed as the number of occurrences of COL1 according to the number of TGFβ1 stimuli: **(A)** 2, **(B)** 4, **(C)** 8, **(D)** 16, **(E)** 32, the periodicity which varies from 22.5 hours to 60 days and the concentration of TGFβ1: 10 (Red), 100 (Blue), 1,000 (Purple), 10,000 (Green) and 100,000 (Yellow) molecules per cell. The first stimulation occurs at day 4. Simulations are expressed as the mean of 5 iterations. P, Periodicity.

## Model predictions validated by new experimental data in mice

To evaluate the model predictions, we analysed COL1 accumulation in a mouse model of CCl4 induced liver fibrosis. For the quantification of collagen deposits, we used second harmonic generation (SHG) microscopy which we have previously demonstrated to quantify

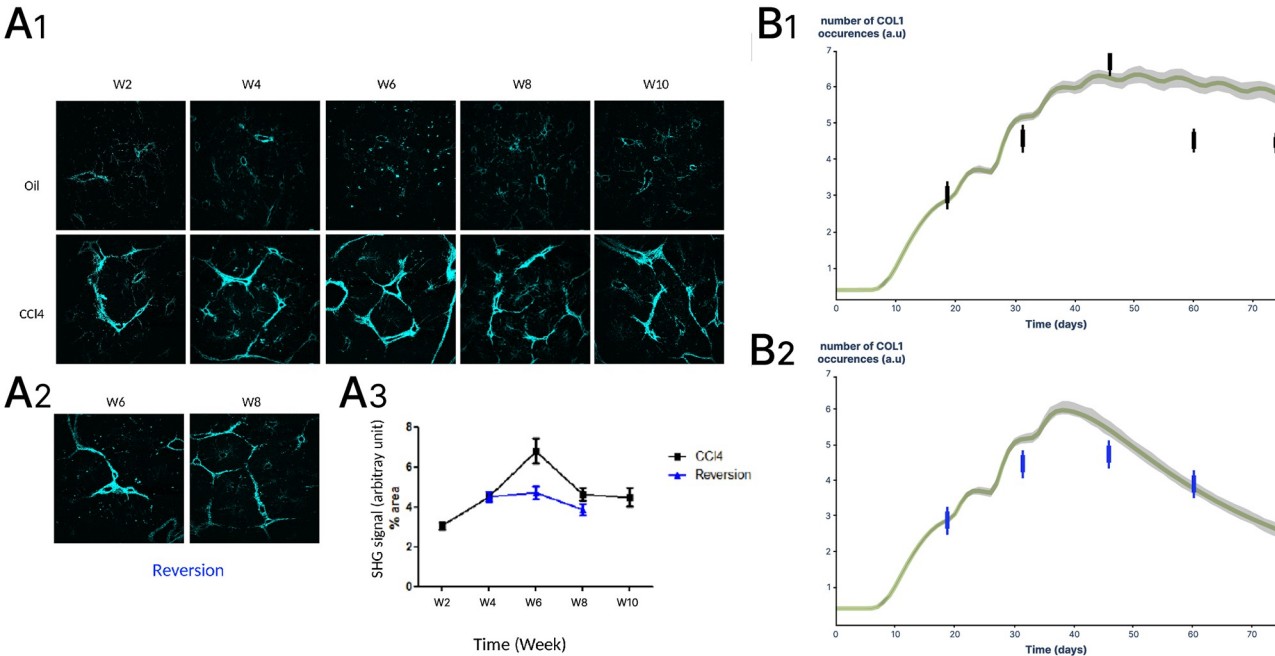

**Fig 4. Comparison of model predictions and experimental data obtained in a mouse model of CCl4 induced liver fibrosis. A1** Representative SHG microscopy images of collagen in mouse livers after 2, 4, 8 and 10 weeks of CCl4 treatment. **A2** For the reversion experiments, treatment was stopped at week 4. **A3** Collagen quantification was performed with ImageJ and results are expressed as percentage of SHG signal relative to total area (Mean ± SD). **B1**, Model simulations were performed using parameters corresponding to the CCl4 injection protocol (three TGF$\beta$1 stimuli in the first week and one stimulus per week for ten weeks). Collagen accumulation was plotted and experimental data (from A1) are indicated with black bars. **B2**, Model simulations were performed using parameters corresponding to the reversion protocol (three TGF$\beta$1 stimuli in the first week and one stimulus per week for 4 weeks). Collagen accumulation was plotted and experimental data (from A1) are indicated with blue bars.

fibrillar collagen in human [32] and mouse [33] livers. Note that the global evaluation of collagen deposits using Sirius red [24] and SHG microscopy may overestimate the contribution of HSCs since other cells can contribute to liver fibrosis such as portal fibroblasts and bone marrow-derived cells [34]. However, HSCs have recently been shown to be the exclusive source of myofibroblasts in CCl4-treated liver [35] which is why we chose data from CCl4-induced liver fibrosis to calibrate our model. As shown in Fig 4A, we compared the signal quantification data obtained by SHG microscopy in our experimental model with the simulation data. Our experimental data showed maximum accumulation of collagen after 6 weeks and stabilization at a lower level at 8 and 10 weeks after treatment (Fig 4A$_1$). In tissue samples from mice that have undergone the reversion protocol, the SHG signal shows a decrease in COL1 at 8 weeks after CCl4 removal (Fig 4A$_2$). For the simulation studies, TGF$\beta$1 parameters were adapted to the experimental protocol, i.e. three TGF$\beta$1 stimuli in the first week and one stimulus per week for ten weeks. CCl4 induces rapid hepatocyte damages leading to inflammation and TGF$\beta$1 production that we modeled using saturating TGF$\beta$1 quantities (10,000 molecules/cell). As shown in Fig 4B$_1$, the predicted curves for COL1 accumulation reached a maximum value similar to the experimental data, then at 8 and 10 weeks, the stabilization plateau was slightly higher in our simulation. Regarding the reversion protocol, the predicted curves for COL1 are in full agreement with the experimental data (Fig 4B$_2$).

We also validated the prediction of COL1 dynamics using experimental data from other laboratories, including liver fibrosis models induced by dimethyl_nitrosamine [36] and thioacetamide [37]. Results are presented in (S2A and S2B Fig).

## iHSC accumulation is associated with human liver fibrosis

A major observation by Kisseleva et al. [24] is that the fibrotic response is exacerbated in mice that have undergone an initial chronic aggression followed by a reversion period allowing a return to a healthy tissue phenotype. This was associated with the presence of inactivated stellate cells (iHSCs) which display an enhanced response to TGF$\beta$1 when stimulated in cell culture, leading to faster and greater production of COL1 [24, 25, 38]. These iHSCs would therefore still be present in the "repaired" liver after six months of reversion. To take this into account, we calibrated our models *reactMFB-with-inactivation* on Kisseleva's observations *et al.* [24] after the second aggression by introducing the hypothesis of a slow elimination of iHSCs. However, there is no experimental data describing the fate of these cells during aggression/reversion cycles. Using the relapse protocol described by Kisseleva *et al.* [24], we studied the behavior of iHSCs throughout injury-repair cycles. This protocol includes chronic injuries induced by eight CCl4 injections over one month, followed by a six-month recovery period, and then a new injury similar to the first. We repeated this cycle three times (Fig 5).

Our simulations show that the dynamics of $\alpha$-SMA expressing cells and amounts of COL1 were consistent with Kisseleva's *et al.* [24] experimental observations, i.e. higher levels of these two markers after the second aggression compared to the first one (Fig 5A and 5B, cycles 1, 2). Importantly, the model allowed us to detail the dynamics of iHSCs that was not experimentally evaluated, and we observed an increased number of iHSCs (Fig 5C, cycles 1, 2). When the aggression/reversion cycles were repeated, we showed an accumulation of iHSCs that was associated with that of $\alpha$-SMA expressing cells and COL1 (Fig 5A, 5B and 5C, cycles 3, 4). The increased number of iHSCs may result from the balance between the higher number of $\alpha$-SMA expressing cells leading to the inactivation of more iHSCs and the half-life of iHSCs. On the basis of these observations, the model predicts a progressive accumulation of iHSCs during cycles of aggression/reversion, suggesting a critical role for iHSCs in the dynamics of fibrosis, with accumulated iHSCs constituting a source of COL1 overproducing cells during reactivation.

To validate this prediction, we searched for iHSC accumulation in samples from patients with liver fibrosis. To this end, we analyzed RNAseq data from a large multicenter study comprising 206 histologically characterized liver samples from patients with non-alcoholic fatty liver disease (NAFLD) including 102 patients with non-alcoholic steatohepatitis (NASH) associated with liver fibrosis [39]. Patients were divided in two groups: with low (F0-F1, n = 34) and high (F3-F4, n = 68) fibrosis grades. To identify iHSCs in human liver samples, we used an iHSC gene expression signature reported by Rosenthal *et al.* [28] in a mouse model of non-alcoholic steatohepatitis (NASH). This signature contained 39 genes that were significantly differentially expressed in iHSCs when compared to both qHSCs and aHSCs. In addition, we used 10 iHSC gene markers identified in a mouse model of CCl4-induced liver fibrosis [24] and validated in an *in vitro* reversion model of human activated cells [38]. The list of these genes is detailed in S1 Table.

As shown in Fig 6A, gene set enrichment analyses (GSEA) using the signature identified by Rosenthal et al. [28] showed that iHSC genes were collectively enriched in the high fibrosis group compared to the low fibrosis group (normalized enriched score = 1.38, p-value = 0.097). We performed a new analysis, adding the 10 marker genes identified by Kisseleva *et al.* and El Taghdouini *et al.* [24, 38] to the list of genes identified by Rosenthal *et al.* [28]. As shown in Fig 6B, this additional set of genes increased the statistical power of the GSEA analysis (normalized enriched score = 1.57, p-value = 0.048), suggesting that the expression of this iHSC-specific gene set is increased in patients with high fibrosis. These observations are in line with the increase in the number of iHSCs predicted by our model, during the progression of hepatic fibrosis.

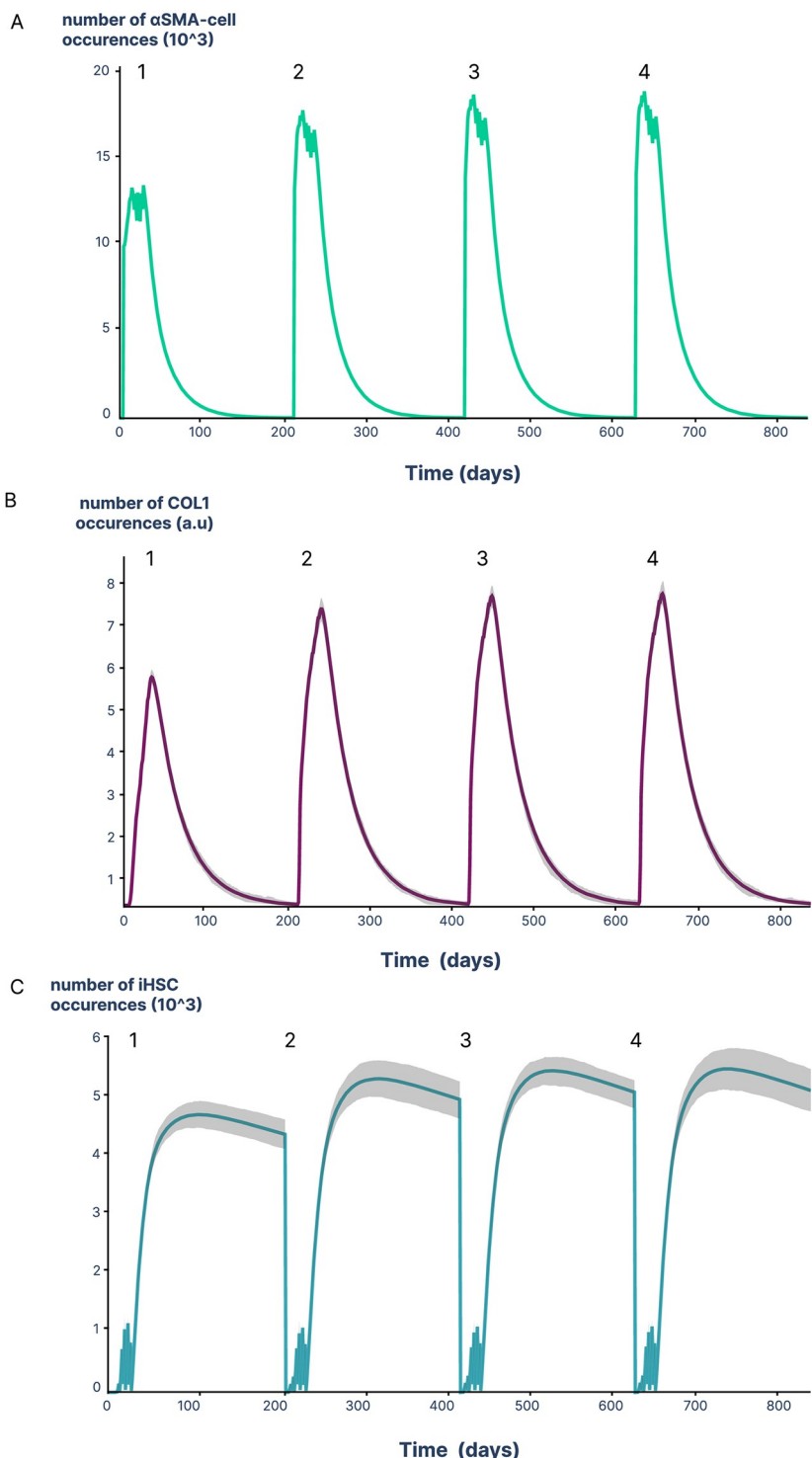

**Fig 5. iHSCs accumulate during injury-repair cycles.** Simulations were performed using repeated cycles comprising a first injury cycle (8 injections twice a week for 1 month), followed by a 6-month recovery period and a second injury cycle as previously described by Kisseleva *et al.* [24]. Simulations are expressed as the mean of 50 replicates for cells (**A**), and all 50 replicates have been represented for COL1 (**B**) and iHSCs (**C**). Overlaps are shown as grey area.

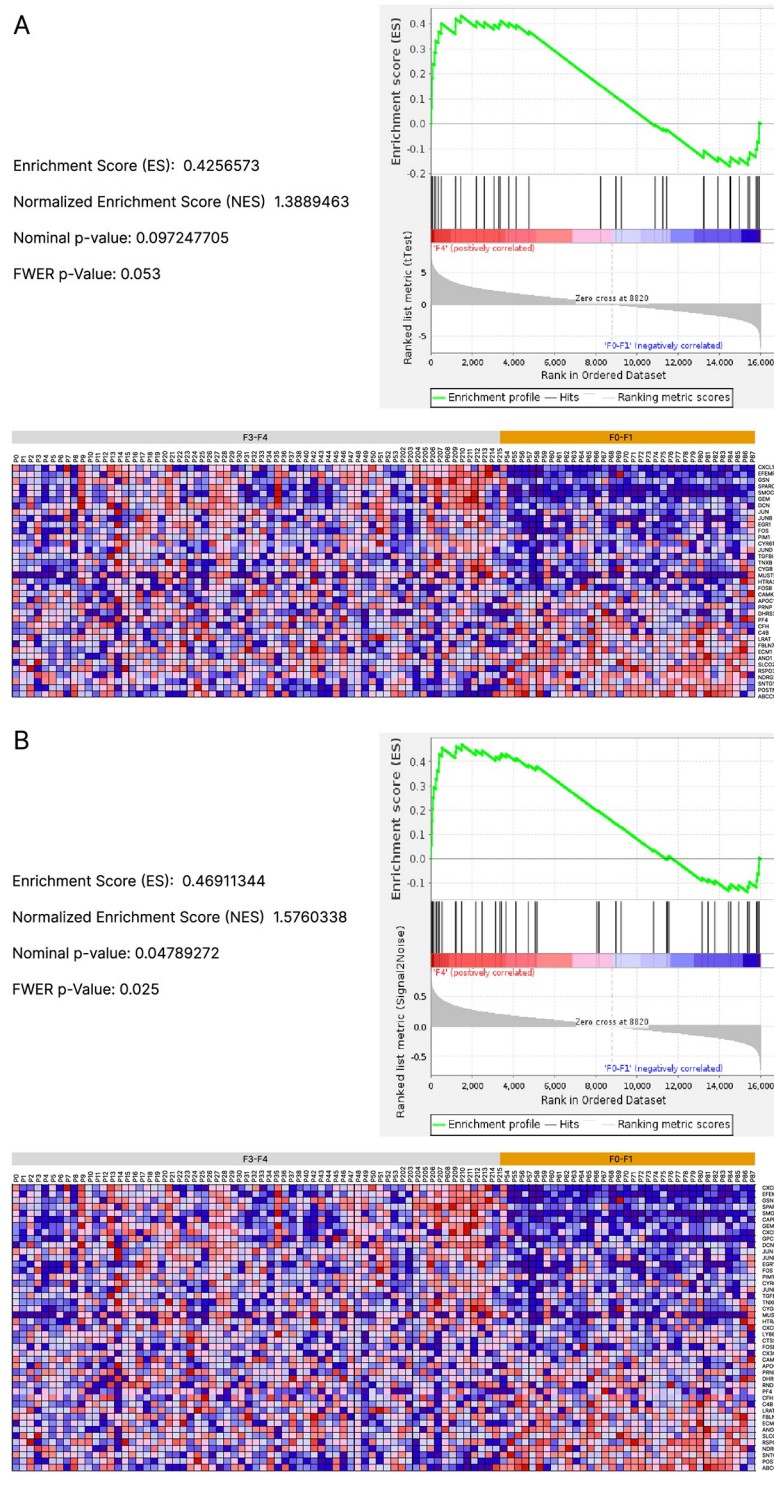

**Fig 6. iHSC accumulation is associated with liver fibrosis.** Gene set enrichment analyses of iHSC gene expression was performed in low fibrosis (F0-F1, n = 34) and high fibrosis (F3-F4, n = 68) samples from patients with non-alcoholic steatohepatitis (NASH) ([39]). **A)** Analysis performed with the iHSC gene signature identified by Rosenthal *et al.* [28]. **B)** Analysis performed with the iHSC gene signature identified by Rosenthal [28], supplemented by ten iHSC markers identified in [24, 38]. Results are presented in the form of correlation profiles between gene lists, enrichment scores (ES) which indicate to which phenotype (low or high fibrosis) the gene set is most positively correlated and heat maps visualizing clusters and gene distribution.

## Discussion

Regardless of the underlying etiology, all chronic liver diseases are associated with the development of hepatic fibrosis, which is a major public health issue. Understanding the dynamics of hepatic fibrosis and its reversion is essential for the clinical management of patients and the adaptation of therapies. A pivotal process in the development of hepatic fibrosis is the TGF$\beta$1-dependent activation of hepatic stellate cells whose plasticity is now widely recognized [40, 41]. Here, we investigated HSC plasticity by developing a multiscale model using a rule-based language. We provided the first model to describe the dynamics of HSCs during liver fibrosis and its reversion, demonstrating the importance of inactivated cells in these processes.

### Kappa, a language adapted to multiscale modelling

While ordinary differential equations (ODEs)-based models have long been used to model biological dynamics, rule-based models (RBM) are much more appropriate for dealing with networks with combinatorial multi-state interactions as discussed by Chylek *et al.* [42]. Indeed, ODE approaches require the implementation of an equation for each species. Agent-based models (ABM), while more appropriate for bottom-up approaches than ODEs, require a description of all possible behaviors of agents. Common ABM approaches are based on the object-oriented paradigm [43] or process calculi [44].

In the case of our model, the single agent has 4 sites, some of which have 8 or 14 different states. Using an ABM approach would have exploded the number of rules ($\sim$ 300) in the model to describe the 39 possible agent states. RBM approaches were therefore the most appropriate to represent the known interactions operating on stellate cells during fibrosis development and reversion. Based on site graph rewriting, BioNetGen [45] and Kappa [21, 22] are the two major languages used for RBM implementation and simulation.

Kappa formalism has been widely used for modeling biological reaction networks such as cell signaling [46], gene regulation [47], epigenetic regulation [48], repressilator system [49] and DNA repair [50]. Using Kappa language, our team previously developed a model to describe the regulatory role of extracellular matrix networks in TGF$\beta$1 activation [51]. The transition from molecular to cell-molecule interaction models has been a new challenge, and we have recently demonstrated the applicability of the Kappa language in multi-scale modeling [52]. However, to overcome the explosion in memory demand due to the declaration of molecules and cells as agents, we have used *tokens* and *counters*. The way *tokens* are stored in memory and the absence of explicit bond with agents considerably reduces the computational cost associated with the existence of different scales in the model. This involves the use of hybrid rules between agent and *token* whose applicability has been demonstrated in our model. *Counters* allowed a more detailed description of the activation and differentiation states of HSCs by creating intermediate steps and cell behaviors depending on their activation/differentiation stage, that better reflects experimental observations. Moreover, *counters* allow to model phenomena that the duration of which is more predictable than chemical interactions. With time-exponentially distributed event, there is only one parameter: standard deviation which is fully characterised by the average time. Erlang distributions provides a mean to narrow the distribution hence leading to events which are more predictable (and which corresponds from a mathematical point of views to the composition of several atomic events following time-exponential distributions).

In short, *counters* and *tokens* are key to multiscale modelling in Kappa. *Tokens* enable a more compact representation of agent populations, *counters* enable a more compact representation of site populations, with both facilitating the search and abstraction of potential rule applications. To our knowledge, beside a model of pulmonary viral infection by William Waites [53], our study is the first one where *counters* and *tokens* have been used in a Kappa model.

## A unique fibrosis model for different etiologies

While the role of TGF$\beta$1 in HSC activation is well established, our model showed how its concentration, the number of stimuli and their periodicity influence the dynamics of liver fibrosis. These observations underline the adaptability of the model to the different pathophysiological conditions associated with the development of fibrosis. In murine models of liver fibrosis induced by chemical compounds such as CCl4, the toxic effect is massive, with strong hepatocyte necrosis, exacerbated inflammatory response and a saturating release of TGF$\beta$1. In other models, such as diet-induced fibrosis, the immune response is different and fibrosis progresses more slowly. Sensitive to small variations in TGF$\beta$1, our model reproduces the collagen accumulation observed in these different fibrosis mouse models. Similarly, the response to liver injury in human is etiology-dependent and may affect the dynamics of HSC activation. In contrast to chemical models, diet-induced fibrosis models developed to mimic the progression of NAFLD to non-alcoholic steatohepatitis (NASH) are characterized by a lower level of fibrosis. In addition, the dynamics of fibrosis are highly variable depending on the rodent strain and diet composition. Moreover, the optimal model of steatohepatitis leading to fibrosis (choline_deficient, amino_acid defined diet) is poorly reversible [54]. To overcome this heterogeneity, we chose a reversible diet model [55] which we compared with model predictions. We set the parameters of TGF$\beta$1 to low levels (4500 molecules/cell, 6 stimuli and a periodicity of 14 days). As shown in supplementary Fig S2-C, the model captured the increase in COL1 but reversion dynamics were too fast, suggesting that for this condition the model requires additional regulators.

## A model to capture stellate cell heterogeneity and microenvironment remodeling

Recent single-cell studies have demonstrated the great heterogeneity of HSCs and MFBs during the development of fibrosis [17, 28, 56]. These works also show that, in addition to a diversity of phenotypes, the dynamics of HSCs during their activation and differentiation process are far from linear, as shown by pseudotime analyses [57, 58]. Our approach manages to model this heterogeneity by introducing intermediate steps in the activation and differentiation process, and allowing cells in each of these states to proliferate, progress through the activation process and produce COL1. The implementation of these rules combined with the SSA algorithm (Stochastic Simulation Algorithm) means that, at each point in the simulation, there are different phenotypes and a 'unique' history is kept for each agent.

However, modelling HSC dynamics does not capture the full complexity of the microenvironment that governs these dynamics such as the multitude of events regulated by immune cells. Other multiscale models include the role of Kupffer cells in HSC activation via TNF-$\alpha$ [13, 15] or antagonistic regulation by M1 and M2 macrophages [16]. Similarly, it is difficult to include all the components involved in extracellular matrix remodeling. This process is finely regulated by the balance between the production of matrix metalloproteinases (MMPs) and their inhibitors, Tissue Inhibitor of metalloproteinases (TIMPs), both secreted by HSCs, MFBs, but also by immune cells involved in the resolution of fibrosis [59–61]. In their model, Friedman & Hao [16] reduced ECM remodeling to the sole contribution of macrophages and did not take into account the known role of HSCs at all. To overcome this molecular and cellular complexity while keeping remodeling process into our own model, we introduced *tokens* to represent low and high remodeling COL1. In this way we modeled both collagen accumulation during TGF$\beta$1 stimulation and its regression upon TGF$\beta$1 withdrawal.

## Inactivation of reactivated MFBs, a critical factor in the progression of fibrosis

The process of inactivating HSC-derived MFBs has been presented as a key element in the reversion of fibrosis enabling the elimination of $\sim$ 50% of MFBs [24, 25]. The authors have shown that the inactivated cells can be reactivated both *in vivo* and *in vitro* in response to a new stimulus, and that they are much more reactive and fibrogenic than initially activated cells [24, 38]. However, the fate of these reactivated cells remained unknown, leading us to hypothesize a potential re-inactivation of these reactivated cells. Our results showed that the inactivation of these cells was essential to maintain COL1 accumulation in chronic lesions, but that this inactivation had to be limited. Indeed, our models have shown that 5% of reactivated MFBs should be inactivated, experimental observations are no longer respected when using higher percentages of reactivated MFBs. This loop of inactivation and reactivation induces a change in the MFB population. As lesions accumulate, cells inactivated between aggressions are reactivated, resulting in a population of reactivated MFBs that becomes the majority and more fibrogenic cells. In support of these predictions, single-cell sequencing data identified these ECM-producing MFB phenotypes with increase fibrogenic activity [28].

## Reversion of inactivated stellate cells to a quiescent state as an alternative to cellular elimination

Although the existence of iHSCs has been supported by several studies, their behavior is not yet well understood. These cells have a phenotype close to that of quiescent cells, but retain the memory of their previous activation [24, 38]. However, the fate of iHSCs in the absence of reactivation stimuli remains unknown. The development of our models enabled us to observe that the reversion of these iHSCs to a quiescent state, regardless of the percentage of cells affected, did not impact the overall behavior of the other components. This may be due mainly to the time required for the elimination of iHSCs and the renewal of quiescent cells, which is particularly slow in our models. This dynamic limits the impact of reversion. It is also possible that these cells have a faster elimination dynamic, but that this is compensated by a proliferation capacity, thus slowing down their elimination and increasing the impact of potential reversion. As it stands, our models suggest that the complete reversion of iHSCs to a quiescent state is a potential means of eliminating them in the absence of TGF$\beta$1. Experimental follow-up of these cells over reversion times of more than one month would provide more precise information on their fate.

## Inactivated HSCs as new markers of fibrosis progression?

The accumulation of iHSCs during fibrosis is a very important concept from a clinical perspective, as the progression of fibrosis is not at all linear. iHSCs within tissues could act as a kind of memory of an initial injury, making the tissue more sensitive to the next injury. Of course, this memory facilitates rapid repair, but the recurrence of these repair episodes, if too close together, accelerates the progression of fibrosis. Although our model is not fully adapted to follow the dynamics of the mouse NASH model due to the slow release of diet-induced TGF$\beta$1, we showed that the observation of iHSC accumulation is supported by the enrichment of iHSC gene signature in NASH patients with increased fibrosis.

Identifying inactivated cells *in vivo* would provide a genuine marker of patient history. However, this identification remains complex due to the intermediate phenotype of these cells, between quiescent and activated state. There is as yet no established phenotypic signature for inactivated cells, and gene expression signatures vary according to the mouse models used

(NASH, CCl4). These phenotypic signatures depend both on the time of reversion and on the methods used to identify them, depending on whether their expression is compared with that of quiescent cells and/or activated cells. Recent advances in single-cell sequencing will enable us to refine these signatures by characterizing more specific groups of genes.

## Conclusion

The present study provides two major findings. First we have demonstrated the applicability of the Kappa language to multi-scale approaches by using: *tokens*, to represent the quantities of molecules in a compact way, and *counters* to abstract the dynamics of cell-specific phenotypes. Second, we developed the first model of HSC dynamics during liver fibrosis and reversion. Thus, our model provides biological predictions for the reactivation loop, the dynamics of inactivated cells and their accumulation during fibrosis progression. These predictions have been validated either experimentally in a mouse model of CCl4-induced fibrosis, or using RNA sequencing data from patients with fibrosis.

## Materials and methods

### Kappa syntax

Kappa is a site graph rewriting language [62–65] that uses a chemistry-inspired syntax to transparently describe the interactions between component occurrences. The syntax used by Kappa is that of site graphs, using rules to describe the behavior of agents over time. An agent describes species (e.g. cells, proteins) and defines the characteristics of these species. Rules define both the interactions between agents and their behavior. Agents are described by one or more *sites* which can have different *states*. These states allow agents to establish links between themselves. A rule describes a process that sets a condition (left-hand member) and an event (right-hand member). All the elements at the left of the $\rightarrow$ are the elements needed for the rule to be applied, the elements at the right are what will be produced. In the following example, two agents A and B interact to form a complex AB.

$$A(x\{p\},\ y[.]),\ B(y[.])\ \rightarrow\ A(x\{p\},\ y[1]),\ B(y[1])\ @\ 'k' \tag{1}$$

The agents A and B and their respective *sites* are written as: A*gent_name*(*site_name*{*state_value*}[*binding_state*]). The agent A has its *site x* in a *state p* and its site *y* free ([.]), and the second agent B has it *site y* free. A and B will bind together via their *site y*. The symbol @ is followed by the expression that defines the rate of each potential application of the rule in the state of the system. That is to say the probability that a potential event effectively occurs within an infinitesimal interval of time.

Simulating models in which agents have to be explicitly bound to other agents can be computationally expensive. Instead, such bounds can be described implicitly. To this aim, the binding *state* of the binding *site* in an agent A can be encoded as an internal state which specifies whether the *site* if free, or bound. In the latter case, the *site* to which this site is bound to is not specified. Then the amount/quantity of agent B remaining is encoded by the means of a *token*. Consequently, *Tokens* constitute a pool of variables enabling the description of abundant molecular species in a continuous manner. Their use allows us to resolve the memory limitation caused by declaring molecular species as an agent. If a *token* such as ATP is added to the rule 1, the following rule is obtained (rule 2), where the binding of A and B requires ATP. What stands between | and @ defines the quantity of *token* to be deleted or added (2) when the rule is applied. Because the binding of A and B depends on the amount of A*TP*, the rate of the

rule has to be modified to include the quantity of $ATP$, written as $|ATP|$.

$$A(x\{p\}, y[.]), B(y[.]) \rightarrow A(x\{p\}, y[1]), B(y[1]) | - 10'ATP'@ |ATP|*'k' \qquad (2)$$

Models in which agents have numerous states require many rules to describe the changes between these states. Instead of describing all these states using numerous rules, it may be sufficient to know the number of steps to follow a process, requiring only a few rules, by increasing the step index. With *counters*, we can use discrete intervals to describe a special type of *site* which can be increased or decreased in the rules. In the following example (rule 3), A has a *site* called *n*, which is a *counter* that counts the number of bonds that can be formed by A during its lifetime in the system. This new rule allows A to bind to B as long as the value of *n* for A is greater or equal to one. When a bond is formed, the value of *n* decreases by one.

$$A(x\{p\}, y[.], n\{\geq 1\}), B(y[.]) \rightarrow A(x\{p\}, y[1], n\{-=1\}), B(y[1]) @ 'k' \qquad (3)$$

However, the current syntax of Kappa cannot allow to test whether the value of a counter is inferior to a given value (i.e., the "$<$" symbol is not supported). To overcome this limitation, it is necessary to use two counters whose sum remains constant. This syntactic limitation is overcome in the new version of Kappa (which internalises the use of pairs of counters to deal with less than inequality tests).

## Modeling HSC dynamics

Using Kappa rules, we described HSC dynamics by including various biological processes such as the activation of quiescent HSCs (qHSC) by TGF$\beta$1 leading to activated HSCs (aHSC), the transdifferentiation of aHSCs into Myofribroblasts (MFB), the elimination of MFBs after removal of TGF$\beta$1 either by apoptosis and senescence pathways (apop_sene_MFBs) or by reverting to an inactivated state (iHSC), the reactivation of iHSCs upon new TGF$\beta$1 stimulation leading to reactivated HSCs (react_HSC) and the transdifferentiation of react_HSCs into reactivated MFBs (react_MFB). We also used rules to describe cell proliferation and production of collagen (except for qHSCs and iHSCs). In these models, HSCs are agents characterized by 3 *sites*: i) the cell state of HSCs (qHSC, aHSC, MFB, iHSC, react_HSC, react_MFB, apop_sene_MFBs), ii) the TGFB1 binding (free or bound) and iii) the state of the TGF$\beta$1 receptor (TGFBR) that can be localized to the membrane, internalized or degraded. We introduced a pair of counters (called intermediary-step and control_counter) to scale the intermediate steps between cell physiological states, and we used 4 *tokens* to describe TGF$\beta$1 and collagen molecules. The resulting families of models contained 75 to 77 rules and 37 to 41 parameters, depending on families. All rules are available in the GitHub repository: https://github.com/MBougueon/HSC_model_Kappa. We detailed here the representative rules of the biological processes. These rules are given in graphic form for easier reading.

**qHSC renewal.** qHSC renewal is modeled using two rules (Fig 7). These rules take into account the fact that the site *TGFB1_binding* on qHSCs must be *free*, since binding of TGF$\beta$1 induces activation into proliferating HSC.

**TGF$\beta$1 binding to cells.** To describe the interaction of TGF$\beta$1 with HSCs, 7 different rules are used according to the cell state. Fig 8 shows the binding rule of TGF$\beta$1 to qHSCs. Only the site *cell_state* of the agent differs between these rules, except for MFBs and react_MFBs, where the rule rate is decreased. The fact that each agent can bind TGF$\beta$1 introduces competition between agents for the binding of TGF$\beta$1.

**Turnover of TGFBR.** The endocytic pathways regulating *TGFBR* receptor signaling and turnover has been previously described by Di Guglielmo *et al.* [66] and modeled by Zi *et al.* [67]. During this process, TGF$\beta$1 binding to TGFBR2 induces the recruitment of TGFBR1,

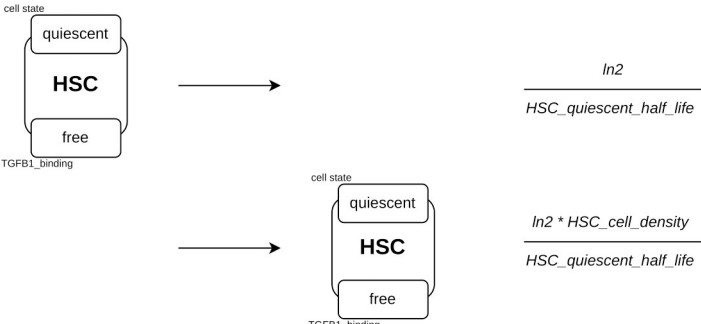

**Fig 7. Renewal of quiescent HSCs.** An agent *HSC* with its site *cell_state* in a state *quiescent* and its site *TGFB1_binding* in a state *free* disappears (degradation). Conversely, the second rule describes the creation of this same agent. The rate of application of each rule is shown on the right.

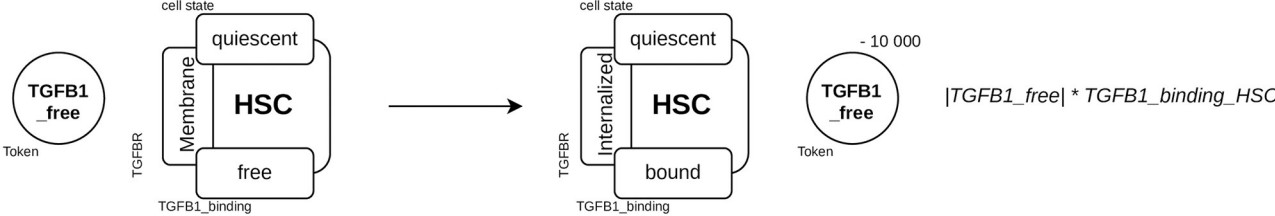

**Fig 8. Binding of TGFβ1.** Binding of TGFβ1 (as a token TGFB1_free) on the site *cell_state* of an *HSC* agent in a *quiescent* state, its *TGFB1_binding* site is in a *free* state and its *TGFBR* site is in a *Membrane* state. When the rule is fired, the value of the token *TGFB1_free* is decreased and the state of the site *TGFB1_binding* changes to *bound* and the state of the site *TGFBR* changes to *Internalized*.

leading to the activation of signaling pathways that regulate TGFβ1-dependent processes such as COL1 expression and cell proliferation. The trafficking of the TGFBR1/TGFBR2 complex between membrane and cytosol takes 30 min and is not affected by TGFβ1 binding. In the present study, we used *TGFBR* to represent the TGFBR1-TGFBR2 complex and we did not describe the intracellular signaling mechanisms. Upon TGFβ1 binding, the receptor *TGFBR* is internalized to transduce signals and can be either recycled (Fig 9A) or degraded (Fig 9B). To

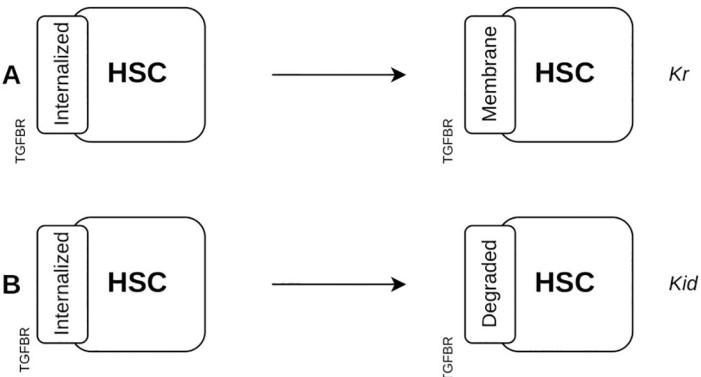

**Fig 9. Degradation and recycling of the TGFβ1 receptor.** (**A**) The *TGFBR* site in an *Internalized* state changes into a *Membrane* state for the recycling rule, (**B**) and into a *Degraded* state for the degradation rule.

**Fig 10. HSC activation process.** The rule specifies that as long as an agent *HSC*, whose site *cell_state* is in the state *activated* has its *counter control_counter* with a value lower than or equal to eight then, the value of its *counter control_counter* will be decremented by one and that of its *counter intermediate_step* will be incremented by one. This simple rule describes the seven stages of activation.

reduce model and time of calculation, the basal turnover of the receptor is not included in the absence of TGF$\beta$1. To model TGFBR synthesis and localization to the membrane, a rule similar to the rule for degradation is used, changing the site *TGFBR* from the state *degraded* to *membrane*.

**HSC activation and differentiation.** During activation and differentiation, cells have the potential to proliferate and produce COL1 (in variable quantities depending on their activation and differentiation state). However, as each agent can only undertake one action at a time, different rules to describe cell activation, differentiation, proliferation and COL1 production are used.

To model HSC activation and differentiation into MFBs, *counters* are used ranging from 0 to 14, corresponding to the 14 days reported in *in vitro* experiments (0 to 7 corresponding to the change from qHSC to aHSC; 8 to 14 corresponding to the changes from aHSC to MFB). On one hand, two rules describe the change of cell state (qHSC to aHSC and aHSC to MFB) and on other hand, two rules describe the step-by-step dynamics by increasing *counter intermediate_step* and decreasing *counter control_counter*.

Increasing the *counter intermediate_step* allows us to follow the activation and differentiation processes. The decrease of the *counter control_counter* ensures that the value of the *counter intermediate_step* remains in the range from 0 to 14 (needed because the "<" symbol is not supported in Kappa). Fig 10 describes the dynamics of the activation, defined in 7 steps, corresponding to the seven days of the transition from qHSC to aHSC, with the implementation of a single rule. For the reactivation process, rules similar to those of the activation process were used.

**Cell proliferation.** A proliferation rule is written for each *counter* value. If an *HSC* agent proliferates, all the sites of the newly created agent will have the same value as those of the parent cell, except for TGFBR. Fig 11 shows an example of proliferation rules for the agent *HSC* whose site *cell_state* is in the *activated* state and its *counter intermediate_step* is equal to 7. The activation or differentiation steps do not affect the proliferation rate of agents and aHSCs and

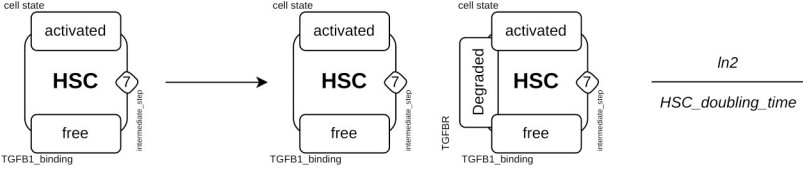

**Fig 11. Example of proliferation rules.** For an agent *HSC* whose site *cell_state* is in the state *activated* and its *counter control_counter* equal to 7.

react_HSCs have additional rules for proliferation upon TGF$\beta$1 binding. There are 30 rules to describe the proliferation process. The value of *state*, *intermediate_step* and *control_counter* is different between each rule to allow all cells to proliferate. Among these rules, 11 allow agents whose site *cell_state* is in the *aHSC* or *react_HSC* states to proliferate upon TGF$\beta$1 stimulation by requiring the site *TGFB1_binding* to be in a state *bound*, then transforming this site into a state *free* after the rule is fired.

**Collagen 1 turnover.** The amount of COL1 in liver fibrosis and reversion results from the balance between production and degradation. Upon TGF$\beta$1 stimulation, activated HSCs and MFBs are the major cell producing COL1 while apop_sene_MFBs contribute to the degradation of COL1 during fibrosis reversion [68, 69]. To overcome the complex molecular mechanisms involved in regulation of COL1 degradation, COL1 was described using two different *tokens*: one for high remodeling (*COL1_remodeling_high*) and another for low remodeling (*COL1_remodeling_low*). These two *tokens* allow the description of the low degradation rate of COL1 during HSC activation and the high degradation rate during the process of reversion. COL1 dynamics is described using 12 rules: 8 for the production of *COL1_remodeling_high* by aHSCs, MFBs, react_HSCs and react_MFBs in the presence and absence of TGF$\beta$1; 2 for the degradation of each *token* (*COL1_remodeling_low* being degraded at a slower rate than *COL1_remodeling_high*) and 2 to enable the two tokens to be linked (Fig 12A). The transformation of *COL1_remodeling_high* to *COL1_remodeling_low* depends on the number of COL1-producing cells (aHSCs, react_HSCs, MFBs and react_MFBs) which are also characterized by expression of $\alpha$-smooth muscle actin ($\alpha$SMA).

The more cell agents are activated (or reactivated) and differentiated into MFBs (or react_MFBs), the greater the production of *COL1_remodeling_high*. A total of 8 rules describe the production of COL1. The state and the variable defining the amount of *COL1_remodeling_high* is modified according to cells (Fig 12B). Of these 8 rules, 4 are TGF$\beta$1-free and 4 are TGF$\beta$1-induced. For the latter 4, the cells must have the site *TGFB1_binding* in a state *bound* and will be transformed into *free* after the production of *COL1_remodeling_high*.

**MFBs inactivation.** After TGF$\beta$1 removal, i.e. in the absence of TGF$\beta$1 stimulation in the model, MFBs and react_MFBs are eliminated through the apoptosis and senescence pathways

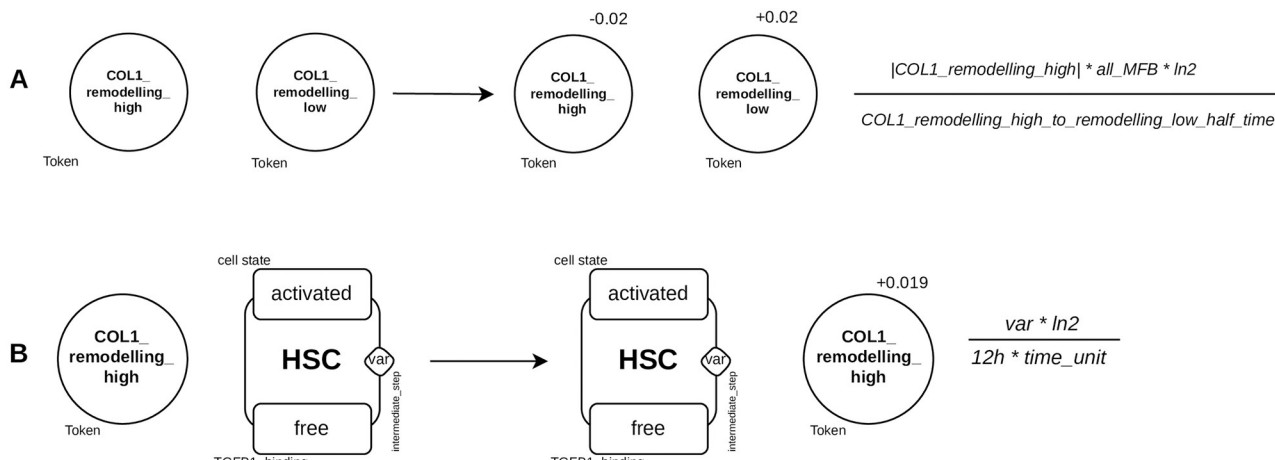

**Fig 12. Example of the rules involved in Collagen 1 turnover. A**), Decrease of collagen degradation, the value of *token COL1_remodeling_high* is decreased and that of *COL1_remodeling_low* is increased; **B**), Production of collagen by activated HSCs, the rate of application of the rule is proportional to the value of *counter intermediate_step*. This enables the cells furthest advanced in the activation process to produce more collagen than those just activated.

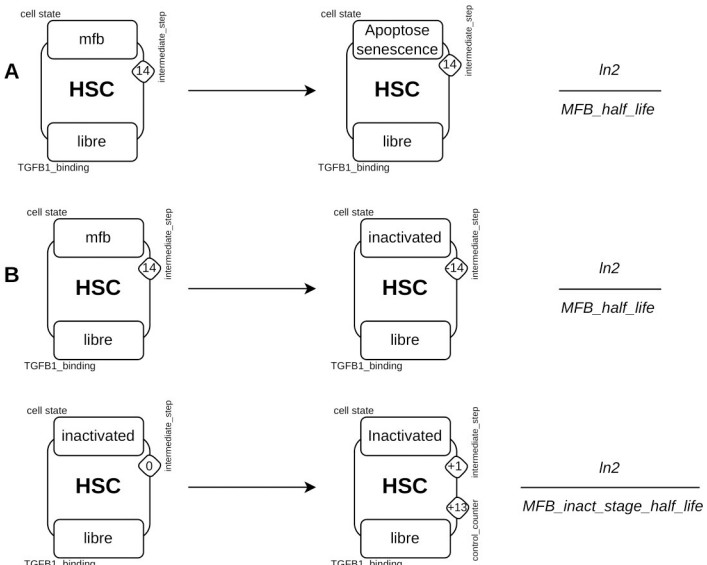

**Fig 13. Rules for eliminating MFBs by apoptosis/senescence and inactivation. A**), The apoptosis/senescence rule consists of changing the site *cellular_state* of an agent *HSC* from a state *MFB* to *apoptosis_senescence*. **B**), For inactivation, two rules are necessary, the first changes the site *cellular_state* of an agent *HSC* from a state *MFB* to *inactivated* while reducing the value of the *counter intermediate_step* by 14. The second rule increases the values of *control_counters* by 13 and *intermediate_step* by 1.

(Fig 13A). An additional rule describes the process of elimination of apop_sene_MFB. The dynamics of the inactivation process is described in two steps using two rules (Fig 13B). The first rule changes the MFBs to an inactivated cell state (iHSC) and decrements the counter value named *intermediate_step* to 0. The second rule increments the value of this same *counters* by 1 and increments the value of the *control_counters* by 13.

## Calibration and parameters estimation

For all the biological processes included in the families of models, 13 parameters were obtained directly from the literature, 9 were calculated and 22 parameters were estimated from biological observations by comparing model simulations with experimental data, running hundreds of simulations (S2 Table). Special attention was paid to calibrate these 22 parameters because of the undocumented information about them, but also because of the interdependence between the parameters and the stochastic approach used.

To calibrate the models, we used a block-based parameter estimation method, i.e. a set of parameters for each subsystem (block) was estimated sequentially. Each set was defined in terms of a specific biological process such as cell proliferation, cell inactivation, collagen production and degradation. Block estimation reduced the impact of interdependence and enabled us to find a set of parameters corresponding to biological observations. As the evaluation of collagen was the experimental model best described, we first calibrated the set of parameters related to on collagen-producing cells. Next, we identified the set of parameters related to *react-MFB inactivation* which strongly affects model dynamics. The third set of parameters focused on iHSC and qHSC dynamics. The final set of parameters focused on collagen itself, encompassing its production and degradation. Following this approach, the estimation of the first set of parameters excluded the family of models called *reactMFB-wo-*

*inactivation*, as shown in Fig 2A. Finally, we performed a global verification to determine whether all the parameters matched the experimental observations (visual inspection).

In addition, the behavior of iHSCs remains unknown, and we developed 3 different families of models taking into account different hypotheses: 1)model family *reactMFB-wo-inactivation* where reactivated MFBs cannot be inactivated, 2) model family *reactMFB-with-inactivation* where reactivated MFBs can be inactivated, 3) model family *iHSC-reversion-to-qHSC* where iHSCs can revert into qHSCs, in the latter model family the react_MFBs can be inactivated. Here, we described the parameters based on biological observations. Note that the rule application rate is drawn according to an exponential distribution whose parameter is equal to the sum of the propensities of all the potential events in the system state. The rates of the rules must therefore be defined as the value $T_{1/2}$ of the biological process they describe.

**Parameters for activation and differentiation rules.** The activation process has been extensively documented using primary cultures of HSCs. Isolated qHSCs are spontaneously activated into aHSCs when cultured on plastic for 7 days, and complete transformation of qHSCs into MFBs takes place within 14 days, with a faster proliferation rate for aHSCs than for MFBs. [70, 71]. Based on these dynamics, the models was calibrated with 15 steps, and the proliferation rate of aHSCs was twice that of MFBs. *In vivo*, HSC activation is mainly driven by TGF$\beta$1 which promotes proliferation and COL1 production. While qHSCs are highly sensitive to TGF$\beta$1, MFBs are much less so [71, 72]. Consequently, we did not introduce rules for TGF$\beta$1-dependent proliferation of MFBs in the models, and TGF$\beta$1-dependent collagen production was five times lower in MFBs than in aHSCs.

**Parameters for TGF$\beta$-dependent rules.** The kinetics of TGF$\beta$1 binding to HSCs have already been reported using *in vitro* cell culture models [73] and a slower binding rate to MFBs has been described due to a decrease in the number of receptors [72]. The parameters of the receptor dynamics are those described by Vilar *et al.* [74] and the half-time required for TGF$\beta$1 to induce a signal initiating the activation process was calculated using the model of Zi *et al.* [67]. This initialization step is necessary to activate the cellular machinery before the activation process begins. Note that this initialization time has been divided by 3.5 for iHSCs, as these cells are much more reactive than qHSCs in response to TGF$\beta$1 [24, 25, 38]. Consistent with this, react_HSCs were allowed to proliferate immediately after being reactivated, whereas proliferation of aHSCs started at the third step of the activation process. This modeling choice was made to simulate the time required for qHSCs to initiate the cellular machinery of activation [75] whereas react_HSCs, due to their activation history, proliferated from the first step of their reactivation.

**Parameters for qHSC self-renewal rate and iHSC elimination rules.** In the absence of information on qHSC self-renewal rate and iHSC half-life, these parameters were estimated on the basis of data reported by Kisseleva *et al.* [24].

**Parameters for MFB and react_MFB elimination rules.** The half-life of MFBs and react_MFBs has not been documented, and was estimated on the basis of observations of the accumulation of these cells during TGF$\beta$1 stimulation and their disappearance 1 to 2 months after the last TGF$\beta$1 stimulation (Kisseleva *et al.* [24]. This accumulation represents an increase in the total number of HSCs after two months of CCl4 treatment from 10.6 to 14.3% of the total liver cell population, where qHSCs and *alpha*-SMA cells account for 10.6% (±0.8) and 14.3% (±1.5), respectively.

**Parameters for collagen 1 dynamic rules.** The rules governing the production, degradation, stabilization, and destabilization of *COL1_high_remodeling* and *COL1_low_remodeling* were calibrated using data from Kisseleva *et al.* [24]. Since COL1 is experimentally quantified as deposits resulting from global remodeling, total COL1 (COL1_tot) is defined as the sum of *COL1_remodeling_high* and *COL1_remodeling_low*. COL1_tot increased 12- and 14-fold after

1 and 2 months of TGF$\beta$1 stimulation (two stimulation per week), respectively. aHSCs and react_HSCs produced 10 time less *COL1_remodeling_high* than MFBs and react_MFBs. However, TGF$\beta$1-dependent COL1 production was increased 10-fold in aHSCs and react_HSCs and only 2-fold in MFBs and react_MFBs, since aHSCs and react_HSCs are more sensitive to TGF$\beta$1 [24, 25, 38]. In the absence of TGF$\beta$1 stimulation, the models were calibrated to observe a rapid decrease in COL1_tot over the first month, followed by a gradual return to its initial value around 6 months later.

**Parameters for the model families *reactMFB-wo-inactivation* and *reactMFB-with-inactivation*.** Upon removal of liver injury, *i.e.* in the absence of TGF$\beta$1 stimulation in the models, 45% of MFBs are inactivated and the remaining 55% are eliminated through apoptosis and senescent pathways [25, 76]. iHSCs can be reactivated upon TGF$\beta$1 stimulation to produce react_MFBs, but no information is available on their fate. Troeger *et al.* hypothesized that react_MFBs can undergo new inactivation/reactivation cycles, but should be limited for each cell [25]. Based on this hypothesis, two model families were created, including model family *reactMFB-wo-inactivation* in which react_MFBs cannot be inactivated and model family *reactMFB-with-inactivation* in which inactivation is possible. For the latter, we developed models by varying the percentage of reactivated MFBs that could be inactivated from 1 to 50%.

**Parameters for the model *iHSC-reversion-to-qHSC*.** Using mouse models of liver fibrosis induced by CCl4, Kisseleva *et al.* [24] assessed the amount of iHSCs compared with qHSCs after removal of CCl4. They reported that qHSCs and iHSCs were in equal proportion after one month of recovery, and that the sum of qHSCs and iHSCs was equal to the initial number of qHSCs after six months of recovery. However, there is no information about the behavior of iHSCs in absence of CCl4 (i.e. of TGF$\beta$1 in the models). In the absence of information on how iHSCs behave in the absence of CCl4 (i.e. TGF$\beta$1 in the models), we developed a family of models with a possible return to the quiescent state by varying the ratio of iHSC that transform into qHSC from 0 to 100%.

## Experimental validation using a mouse model of CCl4-induced liver fibrosis

To validate the predictions of the model, collagen I was quantified during the course of fibrosis and reversion using a mouse model of CCl4 induced liver fibrosis. As previously described [77], seven-week-old female C57Bl/6 mice were treated with oral administration of CCl4 (Sigma-Aldrich, St. Louis, MO, USA) diluted in olive oil. A first dose of 2.4 g/kg of mouse weight was administered to mice three days before starting weekly treatment with a 1.6 g/kg dose for ten weeks. Control mice were treated with the vehicle only (olive oil). Mice were sacrificed at 24 hrs after the last CCl4 dose. For fibrosis reversal experiments, mice were injected for 4 weeks with CCl4 and sacrificed at 8 days (1 week of recovery), or 15 days (2 weeks of recovery) after the last injection. Each group (control, fibrosis and reversion) at each time of sacrifice contained 5 mice. Collagen quantification was performed by SHG microscopy on 20 $\mu$m frozen tissue sections as previously described [78].

## Computational validation using Gene set enrichment analysis of RNA-Seq data from patients with liver fibrosis

The enrichment for iHSC gene expression signature in human fibrotic samples was performed using Gene Set Enrichment Analysis (GSEA) [79, 80]. We used the iHSC gene expression signature previously identified by Rosenthal *et al.* [28] in a mouse model of non-alcoholic steatohepatitis (NASH) and we added iHSC markers previously identified in a mouse model of CCl4 induced liver fibrosis [24] and an *in vitro* reversion model of human activated HSCs [38]. The

list of genes is detailed in supplementary S1 Table. For clinical samples, RNAseq data from patients with non-alcoholic fatty liver disease (NAFLD) [39] were used. Note that three genes (KRT20, GABRA3 and GSTT1) were not identified in the RNAseq data due to a low expression level. Liver samples with fibrosis were selected and separated in two groupss: with low (F0-F1, n = 34) and high (F3-F4, n = 68) fibrosis grade. RNAseq data were filtered using the BIOMART database [81] to select genes encoding proteins and data were normalized using DESeq2 [82] with a threshold of 100 reads. The filtration steps reduced the number of genes from 21,595 to 16,021.

## Supporting information

**S1 Fig. Temporal analysis of cells and COL1 in models with or without reversion of iHSC to a quiescent state.** Simulation series were performed using conditions of stimulation from Kisseleva *et al.* [24]. TGFβ1 parameters were as follows 10,000 molecules per cell, 16 stimulation (twice a week) during 2 months. **(A1)**, models *reactMFB-with-inactivation* with 0% of the iHSCs reverting into qHSC, **(A2)** and **(A3)**, models *iHSC-reversion-to-qHSC* with 50% and 100% of the iHSCs reverting into qHSC, the remaining iHSCs being eliminated. **(B1, B2 and B3)**, variation in the number of cell occurrences for qHSCs, aHSCs, MFBs, iHSCs, react_HSCs, react_MFBs and α-SMA positive cells in the three families of models. The number of α-SMA cells is the sum of the number of aHSCs, MFBs, react_HSCs and react_MFBs. Simulations are expressed as the mean of 10 replicates. **(C1, C2 and C3)**, variation in the number of COL1 occurrence in the three models. Data are expressed as arbitrary unit (a.u) and all 5 replicates are represented.
(TIF)

**S2 Fig. Comparison of model predictions and experimental data acquired in models of fibrosis induced by dimethyl_nitrosamine, thioacetamide and a high-fat diet.** Simulations were carried out using the stimulation protocols described in the different models. **(A)** dimethyl_nitrosamine model: 3 TGFβ1 stimuli per week for 6 weeks [36], **(B)** thioacetamide model: 2 simulations with TGFβ1 per week for 10 weeks [37], **(C)** high-fat model: 6 simulations with TGFβ1 every 14 days for 84 days, adapted to fit Farooq *et al.* results [55]. Collagen accumulation was plotted and experimental data from [36, 37, 55] are indicated with black cross.
(TIF)

**S1 Table. List of the genes used for the Gene Set Enrichment Analysis.** The first column contains the genes from Rosenthal *et al.* It includes 39 genes that are differentially expressed in inactivated stellate cells compared with activated stellate cells and quiescent stellate cells [28]. The second list contains the eight marker genes of inactivated stellate cells identified both in a mouse model of CCl4-induced liver fibrosis [24] and in a *in vitro* reversion model of human activated cells [38].
(XLSX)

**S2 Table. Table of parameters.** *A.U*: Arbitrary Unit corresponding to % areas of COL1 deposits, *Calculated*: parameter calculated directly from values found in the bibliography. *Estimated from biological observations*: parameters for which the values are estimated empirically in order to adapt the families of models to observations at a specific time. All the parameters describing a duration are calculated as the half-time of their reactions. Using the exponential law, the average dynamics of reactions corresponding to the experimental observations was ensured [83] using Block based parameter estimation.
(XLSX)

## Acknowledgments

The authors thank the excellent support of the GenOuest bioinformatics core facility, Dr G Baffet and F Ezan for biological experiments and Dr C Lavau (Inserm U1085, University of Rennes) for proofreading the manuscript.

## Author Contributions

**Conceptualization:** Matthieu Bouguéon, Vincent Legagneux, Octave Hazard, Jérôme Feret, Nathalie Théret.

**Data curation:** Matthieu Bouguéon, Vincent Legagneux, Jérôme Feret, Nathalie Théret.

**Formal analysis:** Matthieu Bouguéon, Jérôme Feret, Nathalie Théret.

**Funding acquisition:** Anne Siegel, Jérôme Feret, Nathalie Théret.

**Investigation:** Matthieu Bouguéon, Jérémy Bomo, Nathalie Théret.

**Methodology:** Matthieu Bouguéon, Vincent Legagneux, Jérôme Feret, Nathalie Théret.

**Project administration:** Anne Siegel, Jérôme Feret, Nathalie Théret.

**Resources:** Nathalie Théret.

**Software:** Matthieu Bouguéon.

**Supervision:** Anne Siegel, Jérôme Feret, Nathalie Théret.

**Validation:** Matthieu Bouguéon, Vincent Legagneux, Nathalie Théret.

**Visualization:** Matthieu Bouguéon, Vincent Legagneux, Nathalie Théret.

**Writing – original draft:** Matthieu Bouguéon, Nathalie Théret.

**Writing – review & editing:** Matthieu Bouguéon, Vincent Legagneux, Anne Siegel, Jérôme Feret, Nathalie Théret.

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
