## [Decision Letter · Decision Letter 0]

11 Apr 2024

Dear Dr. Théret,

Thank you very much for submitting your manuscript "A rule-based multiscale model of hepatic stellate cell plasticity:

critical role of the inactivation loop in fibrosis progression" for consideration at PLOS Computational Biology. As with all papers reviewed by the journal, your manuscript was reviewed by members of the editorial board and by several independent reviewers. The reviewers appreciated the attention to an important topic. Based on the reviews, we are likely to accept this manuscript for publication, providing that you modify the manuscript according to the review recommendations.

Sincerely,

Philip K Maini

Academic Editor

PLOS Computational Biology

Mark Alber

Section Editor

PLOS Computational Biology

Reviewer's Responses to Questions

**Comments to the Authors:**

Reviewer #1: I am not qualified to assess the biological subject matter (I do not know much about livers) so I will confine my comments to the methodology which is sound and a crisp illustration of two mechanisms available in the Kappa language. Most of my comments relate to clarity of exposition which the authors may take or leave at their discretion. One criticism, about parameter estimation, is more substantive, and there is one error about a citation in the text.

The authors present three models, HSC_dynamics_model, HSC_dynamics_model_iHSC_reversion, HSC_dynamics_model_without_reactMFB_inactivation as named in the source code repository (they are called something different in the text -- we may want better naming consistency).

It might be a good idea to explicitly describe their relationship in the main text. Rather than saying "we developed three models ... this model has 75 rules, that model has 76 rules and the other model has 77 rules" (page 4, line 104 onwards), to say something along the lines of, "to get 'reversion' from 'inactivation' we change the rate of one rule and add one extra rule". This would give a better sense of how far in the landscape of models they are from one another and that the changes are really very local. To be fair, Figure 1 does convey this sense but the text could give a different impression (this impression might be an artefact of silly journal rules that say for review purposes, the figures should be as far away as possible from where they belong)

The introduction to the Kappa syntax explains about bonds. However, I think this misses the point somewhat. We could imagine a model without tokens where agents A and B had to explicitly bond to one another. This would be computationally expensive so instead of using a bond, we can say that an agent A bonds to some B but we do not care which B. It is enough simply track the bound state and the number of available Bs. No bonds are used in these models and they are not necessary. Using examples (eqs 2-3) that involve explicit bond formation obscures what I believe is a central point that you are trying to make. Having explained about the use of tokens to avoid the expense of explicit bonds, then explain counters and what, in general terms, they are used for here. I suggest rewriting this section to make it more clearly illustrate these mechanisms as used in these models.

In the models there is an idiom used of a counter called "intermediate_step" together with another called "control_counter". Whilst biologically irrelevant it is necessary for the functioning of the model. Perhaps it would be useful to explain why this is necessary as it would be puzzling for someone unfamiliar with this idiom who tries to read the model. A better soloution would be to modify the Kappa language so that this is not necessary.

Some model parameters come from the literature, some are sensibly calculated and some are fitted. I found the discussion of how they were fitted to be lacking. "22 parameters were estimated from biological data ... numerous simulations before finding the correct set of parameters". This is quite a strong claim for such a large number of free parameters especially given the acknowledged "interdependence between the parameters". How do you know these are "correct" as opposed to one of many possible choices that are consistent with the biological data? The precise method used is left rather vague. I think we need more detail here.

In the discussion of parametrisation, several parameters are swept. This is fine, but the language used is of the form, "we developed models by varying the parameters", suggesting that changing parameters means different models. That is, there are not three but an unbounded number of models! This is a subtle point and might seem minor, and we could talk about parametrised families of models so really what we have here is three families. This is a perennial problem in much of the literature where the word "model" is used to mean many different but related things. I think the paper would benefit from setting a good example by making the language more precise.

A rule-based model using counters by Waites et al is described as unpublished and cited in a footnote. That model (with a slightly incorrect URL, LaTeX appears to have swallowed the ~ character) is in fact the supporting material for this published paper https://royalsocietypublishing.org/doi/10.1098/rsta.2021.0307.

Reviewer #2: The authors present a rule-based model of liver fibrosis focused on the role of hepatic stellate cells (HSC). The model presented in this manuscript builds upon prior agent- and equation-based models of liver fibrosis by including HSC explicitly. The behavior of these cells is fairly granular and includes various states of activation. This model, which was partially validated against data from mice undergoing liver fibrosis secondary to CCl4 administration, suggests a role for TGF-β1-related parameters. Furthermore, the model was used to predict features of chronic liver disease in humans.

While the paper is generally well-written and the modeling work is sound, one key issue needs to be addressed by the authors. This relates to the human validation data (Fig. 6 and related studies). Specifically, this concerns the data on iHSC gene expression. In this analysis, the authors show a ~1.4-fold elevation in iHSC gene expression, with a p value of 0.097. In general, gene expression fold changes below 2-fold, and with p value above 0.05, are not considered relevant. The addition of several genes pushed the fold change in expression to ~1.6 (still less that twofold, albeit with p < 0.05). Thus, these data would be interpreted most conservatively as not showing any major difference, and thus the statement that “….these observations suggest the presence of an increased number of iHSCs during the progression of liver fibrosis, validating the predictions of (the) model,” and similar statements in the Discussion, should be tempered to reflect these relatively weak data. I would suggest noting that these data do not disprove model predictions, but that more data are needed to fully support that hypothesis.

**Have the authors made all data and (if applicable) computational code underlying the findings in their manuscript fully available?**

Reviewer #1: **No: **Whilst the model codes are available, the data used for parameter estimation and the software used for calibration is not provided. It should be unless there is a good reason why it cannot be.

Reviewer #2: Yes

PLOS authors have the option to publish the peer review history of their article (what does this mean?). If published, this will include your full peer review and any attached files.

Reviewer #1: **Yes: **William Waites

Reviewer #2: **Yes: **Yoram Vodovotz

Figure Files:

Data Requirements:

Reproducibility:

References:

---

## [Decision Letter · Decision Letter 1]

5 Jul 2024

Dear Dr. Théret,

We are pleased to inform you that your manuscript 'A rule-based multiscale model of hepatic stellate cell plasticity:

critical role of the inactivation loop in fibrosis progression' has been provisionally accepted for publication in PLOS Computational Biology.

Best regards,

Philip K Maini

Academic Editor

PLOS Computational Biology

Christoph Kaleta

Section Editor

PLOS Computational Biology

Reviewer's Responses to Questions

**Comments to the Authors:**

Reviewer #2: Thank you for addressing my salient concerns.

Reviewer #3: Dear authors,

firstly, I would like to commend you on the excellent work you have done on this manuscript. It is clear that a significant amount of effort and dedication has gone into this research.

I am pleased to see that all the comments from the previous reviewers have been thoroughly addressed in the revised version of the manuscript. This not only shows your commitment to the research but also your respect for the peer-review process.

While my expertise does not lie in agent-based models, rule based models, and Kappa language, my research is primarily focused on the complex phenomenon of fibrosis. I found your model-based hypothesis proposing iHSC as potential biomarkers for fibrosis to be particularly interesting. This concept could potentially be of great value to the research community.

The manuscript is well-written and easy to read, which is a testament to your ability to communicate complex scientific ideas in a clear and concise manner.

Based on the above, I do not have any further comments on your work.

Best regards.

**Have the authors made all data and (if applicable) computational code underlying the findings in their manuscript fully available?**

Reviewer #2: Yes

Reviewer #3: Yes

PLOS authors have the option to publish the peer review history of their article (what does this mean?). If published, this will include your full peer review and any attached files.

Reviewer #2: **Yes: **Yoram Vodovotz

Reviewer #3: **Yes: **Mario Giorgi

---

## [Editor Report · Acceptance letter]

23 Jul 2024

PCOMPBIOL-D-24-00137R1 

A rule-based multiscale model of hepatic stellate cell plasticity:
critical role of the inactivation loop in fibrosis progression

Dear Dr Théret,

I am pleased to inform you that your manuscript has been formally accepted for publication in PLOS Computational Biology. Your manuscript is now with our production department and you will be notified of the publication date in due course.

With kind regards,

Anita Estes
